



# UniFHy v0.1: A community framework for the terrestrial water cycle in Python

Thibault Hallouin[1,2], Richard J. Ellis[3], Douglas B. Clark[3], Simon J. Dadson[3,4], Andrew G. Hughes[5], Bryan N. Lawrence[1,2,6], Grenville M. S. Lister[1,2], and Jan Polcher[7]

[1]National Centre for Atmospheric Science, UK
[2]Department of Meteorology, University of Reading, UK
[3]UK Centre for Ecology & Hydrology, UK
[4]School of Geography and the Environment, University of Oxford, UK
[5]British Geological Survey, UK
[6]Department of Computer Science, University of Reading, UK
[7]Laboratoire de Météorologie Dynamique, IPSL, CNRS, France

**Correspondence:** Thibault Hallouin (th.thibault.hallouin@gmail.com)

**Abstract.** Land surface, hydrological, and groundwater modelling communities all have expertise in simulating the hydrological processes at play in the land system, but these communities have largely remained distinct with limited collaboration between disciplines. In order to address key societal questions regarding the future availability of water resources and the intensity of extreme events such as floods and droughts in a changing climate, these communities must build on the strengths of one

another. The development of a common modelling infrastructure, a framework, can contribute to stimulating cross-fertilisation between them. By allowing (parts of) their existing models to be coupled together, improved land system models can be built to better understand and simulate the terrestrial hydrological cycle. This paper presents a Python implementation of such a framework named the Unified Framework for Hydrology (`unifhy`). The framework aims to provide the technical infrastructure required to couple models, taking into account the specific needs of a land system model. Its conceptual design and technical

capabilities are outlined first, before its usage and useful characteristics are demonstrated through case studies. The limitations of the current framework and necessary future developments are finally presented as a road map for later versions and/or other implementations of the framework.

## 1 Introduction

The Earth's atmosphere and land surface are deeply interconnected systems. Given this, hydrological knowledge is as critical to

atmospheric scientists as meteorological knowledge is to hydrologists. Land surface models have historically been developed as a lower boundary condition to atmospheric models which, to this day, partially explains the shortcomings in the representation of hydrological processes in land surface models. Indeed, the resolution of the land system coupled with the atmosphere has typically been too coarse to adequately represent the spatial structures of the dominant hydrological processes, while the focus on vertical exchanges between the land and the atmosphere has limited the development of the critical lateral redistribution of

water on and below the ground. To overcome these limitations, a modular representation of the terrestrial water cycle using



interconnected modelling components would provide the flexibility required in the spatial discretisation of the land system while preserving the existing coupling approaches with atmospheric models. In addition, such modularity would contribute to developing and comparing alternative representations of the real-world land system and assessing their impacts on hydrological and atmospheric predictions alike.

The land surface modelling community has developed highly configurable models such as the Joint UK Land Environment Simulator (JULES) (Best et al., 2011; Clark et al., 2011), the Organising Carbon and Hydrology In Dynamic Ecosystems (ORCHIDEE) model (Krinner et al., 2005), and the Community Land Model (Lawrence et al., 2019). However, these models do not allow different parts of the land system to be simulated at different explicit resolutions (except for the runoff routing), nor do they make it possible to substitute part of one model with another part from another model. The community has developed

coupling frameworks for the land system, e.g. Landlab (Hobley et al., 2017) can be used to develop modelling components complying with the Basic Modelling Interface (BMI) (Peckham et al., 2013; Hutton et al., 2020) that can be coupled using the Python Modelling Toolkit (Hutton et al., 2021), although as yet none of the existing land surface models above are built using such technologies.

    The hydrological modelling community has also been developing frameworks to compare different physical processes and/or

conceptualisations of the hydrological behaviour of a catchment. For example, FUSE (Clark et al., 2008), SUPERFLEX (Fenicia et al., 2011), and CMF (Kraft et al., 2011) provide bucket-style building blocks to develop integrated catchment models, while SUMMA (Clark et al., 2015) and Raven (Craig et al., 2020) allow the construction of physically-explicit hydrological models using process equations as the building blocks. However, refactoring existing land surface models using these frameworks is not trivial, and an intermediate level of modularity is required. The Open Modelling Interface (OpenMI) has

been developed to provide an international standard to link hydrological and hydraulic models as components. It has been implemented as a flexible approach for allowing models as components to be linked at runtime (Harpham et al., 2019).

    Such intermediate modularity is already a reality within the Earth system modelling community, where atmosphere, ocean, and land components together simulate the dynamics of the Earth system. The technologies used to combine such modelling components range from integrated coupling frameworks such as ESMF (Collins et al., 2005) or CPL7 (Craig et al., 2012),

where existing modelling components require code refactoring to comply with a set of organising and interfacing requirements, to couplers such as OASIS-MCT (Valcke, 2013; Craig et al., 2017), or YAC (Hanke et al., 2016), where existing modelling components require minimal additions to expose their variables to the coupler. While these two families of frameworks vary in the level of intrusiveness into the existing code, they both offer access to essential functionalities such as I/O, parallelism, flexible spatial discretisation, remapping, and so on. This experience and these technologies ought to be exploited to build

modular frameworks to form a new generation of land system models.

    In this manuscript, a new framework for modelling the hydrological cycle in the land system is described, inspired by the modularity of existing Earth system modelling frameworks. It follows an integrated coupling philosophy featuring three framework components interconnected through standardised interfaces allowing new science components to be embedded into the framework with minimal refactorisation of their interfaces. Section 2 expands on its design principles and implementation

details, Section 3 showcases usage of the framework, Section 4 details how to contribute to the framework with new science





components, Section 5 demonstrates the capabilities of the framework on case studies, and finally Section 6 explores avenues for future developments.

## 2 Description of the framework

### 2.1 Modular water cycle blueprint

Given the dominant spatial structures and temporal scales of the processes involved in the terrestrial water cycle, and the interconnected nature of the land system with the atmosphere and the ocean, a modular blueprint featuring three framework components is chosen (see Figure 1): a surface layer component encapsulating the dynamics of moisture and energy exchanges between the atmosphere and the Earth's surface, which are amongst the fastest processes in the terrestrial water cycle and predominantly uni-directional (i.e. vertical); a subsurface component to address the movement of water through the soil down to

the bedrock, which in comparison tends to be slower and truly tri-directional (i.e. lateral redistribution according to topographic and hydraulic head gradients, and vertical percolation/capillary rise/vegetation uptake); and an open water component for the movement of free water in contact with the atmosphere which is of intermediate speed and predominantly bi-directional along the surface of the Earth towards the seas and oceans. Despite this modularity, each component must be conservative with respect to the quantities in the continuity equations so that these are also conserved across the entire land system.

For existing modelling components to be coupled, outputs from one need to be mapped onto inputs for another: this requires a common bank of defined variables to guarantee that the output of one is semantically equivalent to the input of the other. Moreover, to maximise the chances of finding compatible models, this calls for a common interface between components that skilfully yet pragmatically subdivides the terrestrial water cycle continuum. Indeed, a compromise must be found between allowing flexibility in model construction and maximising the potential for existing models to be incorporated in the framework.

This is why a standard interface between the components of the framework is formulated. This interface is a set of prescribed transfers of information between each pair of components in the blueprint. For instance, the open water component is receiving (i.e. inward transfers) 'direct throughfall flux', 'water evaporation flux from open water', 'surface runoff flux delivered to rivers', and 'net subsurface flux to rivers' while it is sending (i.e. outward transfers) 'open water area fraction' and 'open water surface height' (see Figure 1 and Table 1). These interfaces define the relationship between the framework components.

They were designed considering the existing structure of land surface models, namely JULES (Best et al., 2011; Clark et al., 2011) and ORCHIDEE (Krinner et al., 2005). The information transferred through the interface includes the fluxes necessary to fulfil the continuity equations across the entire land system, as well as the diagnostic quantities characterising the state of components which necessarily condition fluxes in other components.

### 2.2 Integrated coupling approach

A first implementation of this blueprint is developed in Python (Hallouin and Ellis, 2021) as an integrated coupling framework following an object oriented approach (see Figure 2 for a visual overview of the software architecture using the Unified





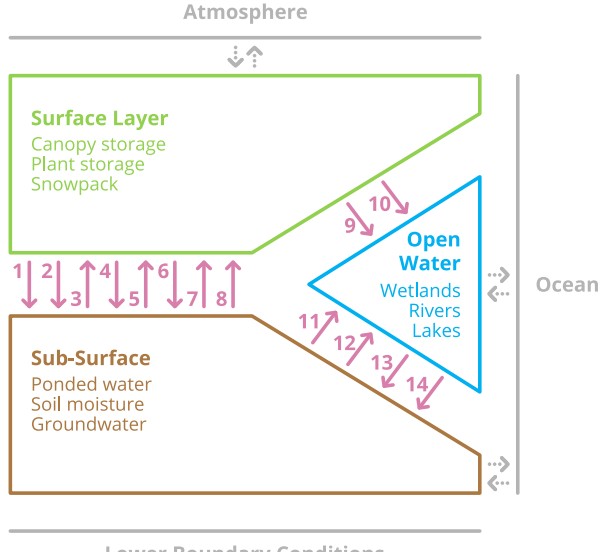

**Figure 1.** Schematic blueprint of the terrestrial water cycle featuring the three components 'Surface Layer', 'Sub-Surface', and 'Open Water', their transfers of information as numbered arrows (see Table 1), and their relationships with external models (atmosphere and ocean).

Modelling Language (UML)). Object oriented programming is ideally suited to efficiently implement such modular software, and the inheritance of the core functionalities of a framework component allows for code reuse in the constitution of the various subdivisions of the water cycle and the community-based component contributions. In this framework, three `Component`
objects are coupled together by a `Model` object and executed concurrently, so that the order in which the components are called does not impact on the outcome of the simulation. The `Model` is responsible for the exchange of information between components, including their potential temporal accumulation and aggregation and/or their potential spatial remapping (using an `Exchanger` object), and is responsible for the time iteration of all components (using a `Clock` object).

The `Component` object provides infrastructure to support the science component; e.g. reading input (using `DataSet` ob-
jects), writing output (using `Record` and `RecordStream` objects), and state memory allocation (using `State` objects). The `Component` class itself is subclassed into the actual framework components represented by the `SurfaceLayerComponent`, `SubSurfaceComponent`, and `OpenWaterComponent` classes, which are used to enforce inward and outward transfers corresponding to the framework interfaces. Each accommodates the description of the physical processes of a given part of the terrestrial water cycle (i.e. the science component) following an initialise-run-finalise paradigm. `DataComponent` and
`NullComponent` classes are also provided as a convenience to allow any of the three framework components to be either replaced with appropriate data or removed. In both cases the replacement generates outward data transfers, in the former case, from data, in the latter case, zeros. Attempted inward data transfers are quietly ignored.





**Table 1.** Prescribed interface variables defining what must be transferred between the framework components.

| # | Name | Unit |
|---|------|------|
| 1 | canopy_liquid_throughfall_and_snow_melt_flux | $\mathrm{kg\,m^{-2}\,s^{-1}}$ |
| 2 | transpiration_flux_from_root_uptake | $\mathrm{kg\,m^{-2}\,s^{-1}}$ |
| 3 | soil_water_stress_for_transpiration | 1 |
| 4 | direct_water_evaporation_flux_from_soil | $\mathrm{kg\,m^{-2}\,s^{-1}}$ |
| 5 | soil_water_stress_for_direct_soil_evaporation | 1 |
| 6 | water_evaporation_flux_from_standing_water | $\mathrm{kg\,m^{-2}\,s^{-1}}$ |
| 7 | standing_water_area_fraction[*] | 1 |
| 8 | total_water_area_fraction[*] | 1 |
| 9 | water_evaporation_flux_from_open_water | $\mathrm{kg\,m^{-2}\,s^{-1}}$ |
| 10 | direct_throughfall_flux | $\mathrm{kg\,m^{-2}\,s^{-1}}$ |
| 11 | surface_runoff_flux_delivered_to_rivers | $\mathrm{kg\,m^{-2}\,s^{-1}}$ |
| 12 | net_groundwater_flux_to_rivers | $\mathrm{kg\,m^{-2}\,s^{-1}}$ |
| 13 | open_water_area_fraction[*] | 1 |
| 14 | open_water_surface_height | m |

[*] Standing and open water both refer to the water on the land surface in direct contact with the atmosphere, but the former corresponds to the ephemeral water on the land surface, while the latter corresponds to the water in rivers and lakes. Total water refers to the combination of standing and open water, taking into account any overlap between the two.

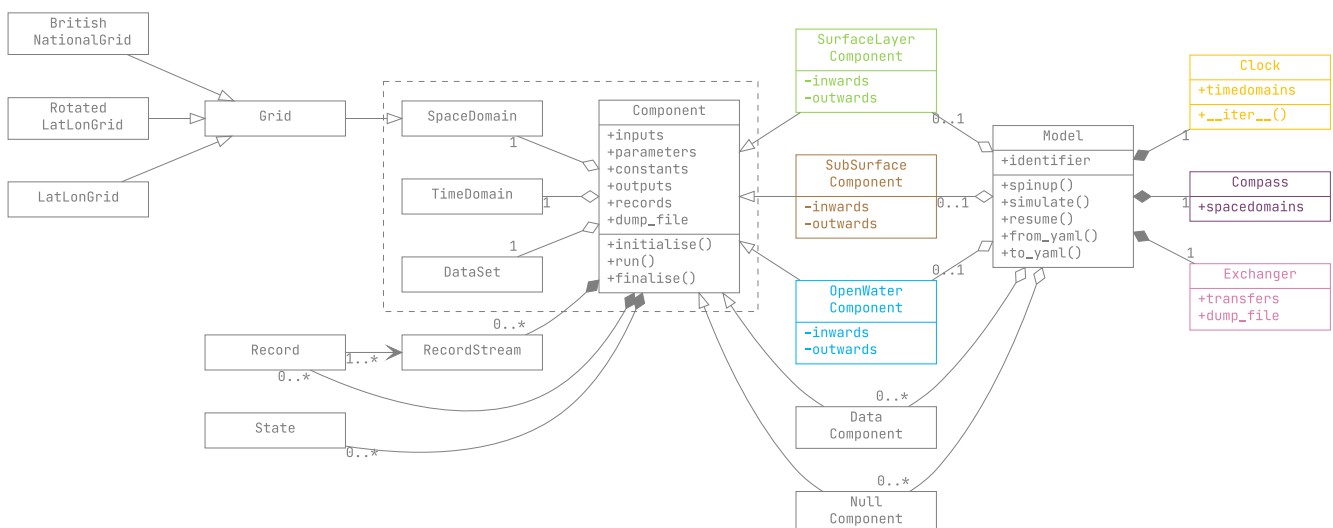

**Figure 2.** UML class diagram of the `unifhy` Python package.



## 2.3 Flexible discretisation

The framework modularity makes it possible to resolve the processes in each science component at their own temporal and spa-
tial resolutions. Each `Component` is discretised individually with its own instances of the `TimeDomain` and `SpaceDomain`
classes defining their temporal and spatial discretisations, respectively.

In the framework, the `TimeDomain` class limits instances to temporal discretisations that are regularly-spaced. While
each component could theoretically run on any temporal resolution independent of the resolution of the other components,
it is essential to make sure that restarting times exist across the simulation period in case of unexpected interruption of the
execution. In order to achieve this, the `Clock` object makes sure that the temporal resolutions are constrained such that, across
the three components, the component temporal resolutions are required to be integer multiples of each other, and the component
temporal extents to span the same simulation period.

In the framework, the `SpaceDomain` class is subclassed into a `Grid`, intended to encompass all structured gridded spatial
discretisations. The distinction between `SpaceDomain` and `Grid` is done in anticipation of additional subclasses to be created
in the future (see subsection 6.2). The `Grid` class itself currently features three subclasses corresponding to two discretisa-
tions on spherical coordinate systems (latitude-longitude and rotated-pole latitude-longitude) as well as one discretisation on
a Cartesian (projected) coordinate system (the British national grid), but additional subclasses can easily be developed. Inter-
nal spatial remapping between differing component discretisations relies on the remapping functionality provided by ESMF
(Collins et al., 2005). If components are to be resolved on different spatial discretisations, not only the components must con-
serve the quantities in the continuity equations, but the remapping operation must also be conservative. Discontinuities being
intrinsic to the land system, e.g. in land cover or soil properties, it appears unrealistic to directly apply traditional interpola-
tion methods for the remapping since they assume continuity, whereas supermeshing techniques (Farrell et al., 2009), where a
supermesh is the union of the components meshes, offer solutions to remain conservative in the remapping without the need
for a continuity assumption. Since the current implementation of the framework does not yet feature an explicit supermeshing
technique across the three components, the `Compass` object makes sure that components are discretised using space domains
of the same class (i.e. in the same coordinate system), that they span the same region, and that their spatial resolutions are
encapsulated in one another, which effectively guarantees that for each pair of coupled components, one is the supermesh for
both of them.

## 2.4 Open science library

Alongside the framework infrastructure itself, an initial library of open source science components complying with the standard
framework interface is available. This allows users to explore alternative combination of components as alternative solutions
to simulate the terrestrial water cycle.

Additional components can be exploited by the framework so long as they comply, or can be made to comply, with the
framework interface via a particular Python class template (see Script 7).

Land system and hydrological modellers are encouraged to become contributors to the framework by sharing their science components with the rest of the community. These contributions can be implemented purely in Python, but can also rely on Fortran, C, or C++ programs called by interface middleware. Contributors need not handle basic functionalities such as memory allocation nor input/output operations, as these are handled by the framework. Ideally, the use of the framework will simplify model development allowing framework contributors to focus on scientific development (see section 4).

## 2.5   Meaningful data

The interface specification is effectively a data specification. To guarantee the unambiguous specification of that interface, as well as to bring to the framework and to users alike a full awareness of the physical meaning and spatio-temporal context of input and output data, the NetCDF Climate and Forecast (CF) Metadata Conventions (Eaton et al., 2020) are exploited in the framework. These conventions provide a robust guide for describing, processing, and sharing geophysical data files. They are

used in a variety of applications, including for global model inter-comparison efforts such as CMIPs (e.g. Eyring et al., 2016).

The `DataSet` class responsible for providing each `Component` with input data relies on reading CF-NetCDF files. This enables the framework to check the compatibility between the data and the configured component, both physically and spatio-temporally. In addition, all record files and dump files are generated as CF-NetCDF files. Such CF-NetCDF files are processed with the package cf-python (Hassell et al., 2017; Hassell and Bartholomew, 2020).

## 3   Usage of the framework

### 3.1   Configuration

In this section, the framework configuration workflow is presented. The user can configure the framework either using the Application Programming Interface (API) directly, or using an intermediate YAML (Yet Another Markup Language) configuration file.

### 3.1.1   Application Programming Interface

The first step in this workflow is to define the temporal and spatial discretisations. The user has to instantiate `TimeDomain` and `SpaceDomain` objects. The framework comes with a variety of constructor methods for these two objects, including using existing data structures, e.g. `from_field`, or using the limits and spacing of the discretisation, e.g. `from_start_end_step` for `TimeDomain` and `from_extent_and_resolution` for `SpaceDomain`. Examples using the latter are presented in

Script 1.

The second step consists in selecting the NetCDF files containing the input data. To do so, the user has to instantiate a `DataSet` object (Script 2). These files must comply with the CF-conventions (subsection 2.5).

The third step, which completes the configuration of a `Component`, is to provide these three objects to the selected component alongside values with units for the component parameters and constants. Additionally, the user can select the variables to





```python
from unifhy.time import TimeDomain
from datetime import datetime, timedelta
from unifhy.space import LatLonGrid

td = TimeDomain.from_start_end_step(
    start=datetime(2007, 1, 1, 0, 0, 0),
    end=datetime(2017, 1, 1, 0, 0, 0),
    step=timedelta(hours=1)
)

sd = LatLonGrid.from_extent_and_resolution(
    latitude_extent=(49, 55),
    latitude_resolution=0.5,
    longitude_extent=(-11, 3),
    longitude_resolution=0.5
)
```

**Script 1.** Example of temporal discretisation (to generate hourly timestepping), and spatial discretisation (to generate a regular 0.5° latitude-longitude grid) using the framework's API.

```python
from unifhy.data import DataSet

ds = DataSet(
    files=['file.nc'],
    select='flow_accumulation'
)
```

**Script 2.** Example of data specification using the framework's API: selecting a variable in a CF-NetCDF file.





record for this component, whether component outward transfers and/or component outputs and/or component states with customisable temporal resolution (multiple of its `TimeDomain` resolution) and summary statistics (mean, minimum, maximum, instantaneous). In Script 3, the open water science component RFM, based on the RFM model (Bell et al., 2007; Dadson et al., 2011), is used and configured. Instantiating a Component will be successful only if all the inputs, parameters, and constants required by the science component are provided and compatible in names, units, time and space dimensions with the component

and its time and space domains.

```
from unifhycontrib.rfm import OpenWaterComponent
ow = OpenWaterComponent(
timedomain=td,
spacedomain=sd,
dataset=ds,
saving_directory='.',
parameters={'c_land': [0.20, 'm s-1'], 'cb_land': [0.10, 'm s-1'],
'c_river': [0.62, 'm s-1'], 'cb_river': [0.15, 'm s-1'],
'ret_l': [0.20, '1'], 'ret_r': [0.10, '1'],
'routing_length': [50000, 'm']},
constants={'a_thres': [1, '1']},
outputs={'outgoing_water_volume_transport_'
'along_river_channel': {timedelta(days=1): ['mean']}
)
```

**Script 3.** Example of component configuration using the framework's API: an open water component based on the RFM model.

The fourth and last step in the configuration workflow is to gather three components, one of each of the three types `SurfaceLayerComponent`, `SubSurfaceComponent`, and `OpenWaterComponent` to form a model. In Script 4, the variables 'sl', 'ss', and 'ow' are instances of each, respectively, configured similarly to the example in Script 3.

Note, the three components forming the model need to comply with the temporal and spatial discretisation constraints

175 formulated in subsection 2.3 for the instantiation of a `Model` to be successful.

### 3.1.2 Configuration file

An alternative to the API is the use of a configuration file written using the human-readable serialisation language YAML. This provides both a more accessible configuration approach for users less comfortable with programming and a way to easily share configurations with other users. The complete configuration workflow presented above using the API can be formulated in a





```
from unifhy.model import Model
md = Model(
identifier='example_run',
config_directory='.',
saving_directory='.',
surfacelayer=sl,
subsurface=ss,
openwater=ow
)
```

**Script 4.** Example of model configuration using the framework's API.

180  single YAML file. For reasons of brevity, only an equivalent for the third step above, i.e. configuring one component (Script 3), is presented in Configuration 1. Entire configuration files can be found in the Supplement.

Configuration files can then be loaded using the API to instantiate the `Model` directly (Script 5). Note, after the successful instantiation of a `Model` using the API (i.e. Script 4), such a YAML file is automatically created in the configuration directory.

```
1   from unifhy.model import Model
2
3   md = Model.from_yaml('my_file.yml')
```

**Script 5.** Example using the framework's API to instantiate a Model from a YAML configuration file.

## 3.2  Simulation

185  A configured `Model` can then be used to start model spin-up cycle(s) and/or to start a simulation run over the entire simulation period specified in the time domains of the components (Script 6). The spin-up period can either be within or outside of the simulation period, so long as the datasets given to the components contain data for it.

Both spin-up and simulation runs can produce dump files, that is files containing intermediate snapshots in the simulation period with all the information required to resume the simulation in case of an unexpected interruption. The user can specify a

190  dumping frequency to choose how often such snapshots should be saved. Once the `Model` is re-instantiated using its configuration file created through Script 4, the simulation can be resumed using any snapshot in these dump files. Moreover, these files can be used to provide initial conditions for the component states in replacement or in addition to the spin-up cycles.





```
openwater:
module: unifhycontrib.rfm
class: OpenWaterComponent
saving_directory: .
timedomain:
start: 2007-01-01 00:00:00
end: 2017-01-01 00:00:00
step: timedelta(hours=1)
spacedomain:
class: LatLonGrid
latitude_extent: [49, 55]
latitude_resolution: 0.5
longitude_extent: [-11, 3]
longitude_resolution: 0.5
dataset:
i_area:
files: [file.nc]
select: flow_accumulation
parameters:
c_land: [0.20, 'm s-1']
cb_land: [0.10, 'm s-1']
c_river: [0.62, 'm s-1']
cb_river: [0.15, 'm s-1']
ret_l: [0.20, '1']
ret_r: [0.10, '1']
routing_length: [50000, 'm']
constants:
a_thres: [1, '1']
records:
outgoing_water_volume_transport_along_river_channel:
timedelta(days=1): [mean]
```

**Configuration 1.** Excerpt from YAML configuration file equivalent to Script 3.





```
from datetime import datetime, timedelta
md.spin_up(start=datetime(2007, 1, 1, 0, 0, 0),
end=datetime(2008, 1, 1, 0, 0, 0),
cycles=3)
md.simulate(dumping_frequency=timedelta(days=100))
```

**Script 6.** Example using the framework's API to start simulations with a Model.

## 4   Contribution to the framework

If the science components already available in the open science library (subsection 2.4) are not sufficient or suitable for the needs of users, they have the opportunity to create their own. New science components for use in the framework can be developed as Python subclasses of the framework's internal `SurfaceLayerComponent`, `SubSurfaceComponent`, and `OpenWaterComponent` classes.

The approach to developing a science component is designed to require minimal development effort, and can be divided into five steps. The first step is to declare a Python class whose base class is one of the `SurfaceLayerComponent`, `SubSurfaceComponent`, or `OpenWaterComponent` classes (e.g. lines 1-3 in Script 7). The second step is to provide a description for the component using the docstring of the class (e.g. line 4 in Script 7). The third step is to declare the component interface, i.e. to indicate which transfers in the standard interface are used and produced (e.g. lines 7-8 in Script 7). The fourth step is to define the component characteristics, including its inputs, parameters, constants, states, and outputs in their corresponding class attributes (e.g. lines 9-23 in Script 7). The fifth and last step is to implement the three class methods `initialise`, `run`, and `finalise` (e.g. lines 26-34 in Script 7, where the `pass` statements should be replaced by the actual implementation of these methods). This initialise-run-finalise (IRF) paradigm is based on the interfacing standards BMI (Peckham et al., 2013; Hutton et al., 2020) and OpenMI (Harpham et al., 2019).

Instances of newly created `Component` classes can then be created and, thanks to their base class, they inherit the functionality that make them readily usable in the framework, as described in section 3.

For existing models, the contributor may need to perform some refactoring of their source code, namely to comply with the framework interfaces and to comply with the initialise-run-finalise paradigm. The creation of a Python class is a requirement for use in the framework, however, the `initialise`, `run`, and `finalise` methods can call software which can be interfaced with Python, such as existing Fortran, C, or C++ programs.

A blank template is available on GitHub at unifhy-org/unifhycontrib-template to provide a starting point for contributors to package their new or existing models into framework compatible Python libraries.



```python
import unifhy

class SurfaceLayerComponent(unifhy.component.SurfaceLayerComponent):
    """Component description"""

    # Component definition
    _inwards = {'inward_name'}
    _outwards = {'outward_name'}
    _inputs_info = {
        'input_name': {'units' : '',  'kind': ''}
    }
    _parameters_info = {
        'parameter_name': {'units' : '', 'valid_range': [0., 1.]}
    }
    _constants_info = {
        'constant_name': {'units' : '',  'default_value': 1.}
    }
    _states_info = {
        'state_name': {'units' : ''}
    }
    _outputs_info = {
        'output_name': {'units' : ''}
    }

    # Component implementation
    def initialise(self, input_name, parameter_name, constant_name, state_name, **kwargs):
        pass

    def run(self, inward_name, input_name, parameter_name,
            constant_name, state_name, **kwargs):
        pass

    def finalise(self, parameter_name, constant_name, state_name, **kwargs):
        pass
```

**Script 7.** Template for a science component contribution: example on a fictional surface layer component.



## 5 Case studies using the framework

### 5.1 Selected science components

A selection of existing models have already been refactored into science components compatible with the framework. These include the Artemis (Dadson et al., 2021), RFM (Lewis and Hallouin, 2021), and SMART (Hallouin et al., 2021) models.

The Artemis model provides a simple runoff production model designed to be comparable with the runoff-production models typically embedded within climate models, which combines Penman-Monteith evaporation (Monteith, 1965) with Rutter-Gash canopy interception (Gash, 1979), TOPMODEL runoff production (Clark and Gedney, 2008), and a degree-day-based snow accumulation and melting model (Moore et al., 1999; Hock, 2003; Beven, 2012). The River Flow Model (RFM) is a runoff routing model based on a discrete approximation of the one-directional kinematic wave with lateral inflow (Bell et al., 2007;

Dadson et al., 2011). The Soil Moisture Accounting and Routing for Transport (SMART) model is a bucket-style rainfall-runoff model based on the soil layers concept (Mockler et al., 2016).

    Note, the Artemis and RFM model parameters are not optimised, while the SMART model parameters are optimised for each catchment separately using a standalone version of the model (Hallouin et al., 2019) and selecting the best performing parameter set from a Latin Hypercube Sampling (McKay et al., 2000) of $10^6$ parameter sets, using a subset for the period

1998-2007 of the driving and observational data used by Smith et al. (2019), and the modified Kling-Gupta Efficiency (Kling et al., 2012) as objective function.

### 5.2 Selected configurations

The capabilities of the framework are demonstrated through three different configurations summarised in Table 2.

    The first configuration puts the Artemis and the RFM models together to form a simple land system model. It demonstrates

the flexibility in the temporal and the spatial resolutions of the various components. Indeed, the surface layer and the subsurface components are taken from the Artemis model and configured to run at an hourly timestep on a 0.5 degree resolution latitude-longitude grid, while the open water component from RFM is used and configured to run at 15-minute intervals on a 0.5/60 (~0.008) degree resolution on a latitude-longitude grid.

    The second configuration demonstrates the possibility to replace science components with datasets. To do so, the surface and

subsurface runoff outputs from the JULES model (Best et al., 2011; Clark et al., 2011) available in the CHESS-land dataset (Martínez-de la Torre et al., 2018) are put together as a `DataComponent` and used in place of the subsurface component, which is then coupled with the open water component of the RFM model, both on a 1 kilometre resolution on the British National grid. The surface layer component is removed by setting it as a `NullComponent`.

    The third and last configuration puts together the Artemis and SMART models. It demonstrates the possibility to substitute

parts of an existing model (i.e. SMART) with parts from another model (i.e. Artemis) and explore the impacts on the model performance. The SMART model is a rainfall-runoff model for application to hydrologically meaningful spatial elements (e.g. catchments, sub-basins), for which the existing gridded space domains are irrelevant. However, the model can be run on a single

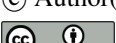



**Table 2.** Framework configurations.

| Framework Component | Science Component | TimeDomain | SpaceDomain | DataSet[1] |
|---|---|---|---|---|
| *Configuration 1: running a simple land system model* | | | | |
| SurfaceLayerComponent | Artemis | Hourly | WGS84 0.5 degree | ERA5 |
| SubSurfaceComponent | Artemis | Hourly | WGS84 0.5 degree | ERA5 |
| OpenWaterComponent | RFM | 15 minutes | WGS84 ~0.008 degree | HydroSHEDS |
| *Configuration 2: routing previously simulated runoff* | | | | |
| SurfaceLayerComponent | NullComponent | – | – | – |
| SubSurfaceComponent | DataComponent | Daily | OSGB 1 km | CHESS-land |
| OpenWaterComponent | RFM | 15 minutes | OSGB 1 km | CHESS-land |
| *Configuration 3: adding explicit surface evaporation processes in a rainfall-runoff model* | | | | |
| SurfaceLayerComponent | Artemis | Hourly | WGS84 0.5 degree | ERA5 |
| SubSurfaceComponent | SMART | Hourly | Catchment (one grid cell) | – |
| OpenWaterComponent | SMART | Hourly | Catchment (one grid cell) | – |

[1] See Appendix A for more details on data sources.

spatial element assumed to represent the whole catchment until more complex geometries are supported in the framework (see subsection 6.2).

Note, the details of the three configurations are available as YAML configuration files in the Supplement.

### 5.3 Selected study catchments

The three configurations are applied to three British catchments, selected to explore the capabilities of the framework: the upper Severn catchment predominantly located in Wales, the Ouse catchment located in North East England, and the Tay catchment located in East Scotland (see Figure 3). These three catchments cover a range of climatological, topographical and geological

settings. Their base flow indices (BFI) are 0.53, 0.39, and 0.64, respectively (Boorman et al., 1995). The three configurations applied to these three study catchments form nine case studies. The simulation period considered is 2008-2017.

### 5.4 Results

Figure 4 showcases the river discharge simulated with the three framework configurations described above, focussing on the river discharge at the catchment outlet in the line plots (a, c, e), and the spatial distribution of river discharge at the end of

the simulation in the gridded plots (b, d). For reasons of brevity, only the Tay catchment is shown in the main text, figures for the other two study catchments are available in Appendix B. These figures confirm qualitatively the plausibility of the



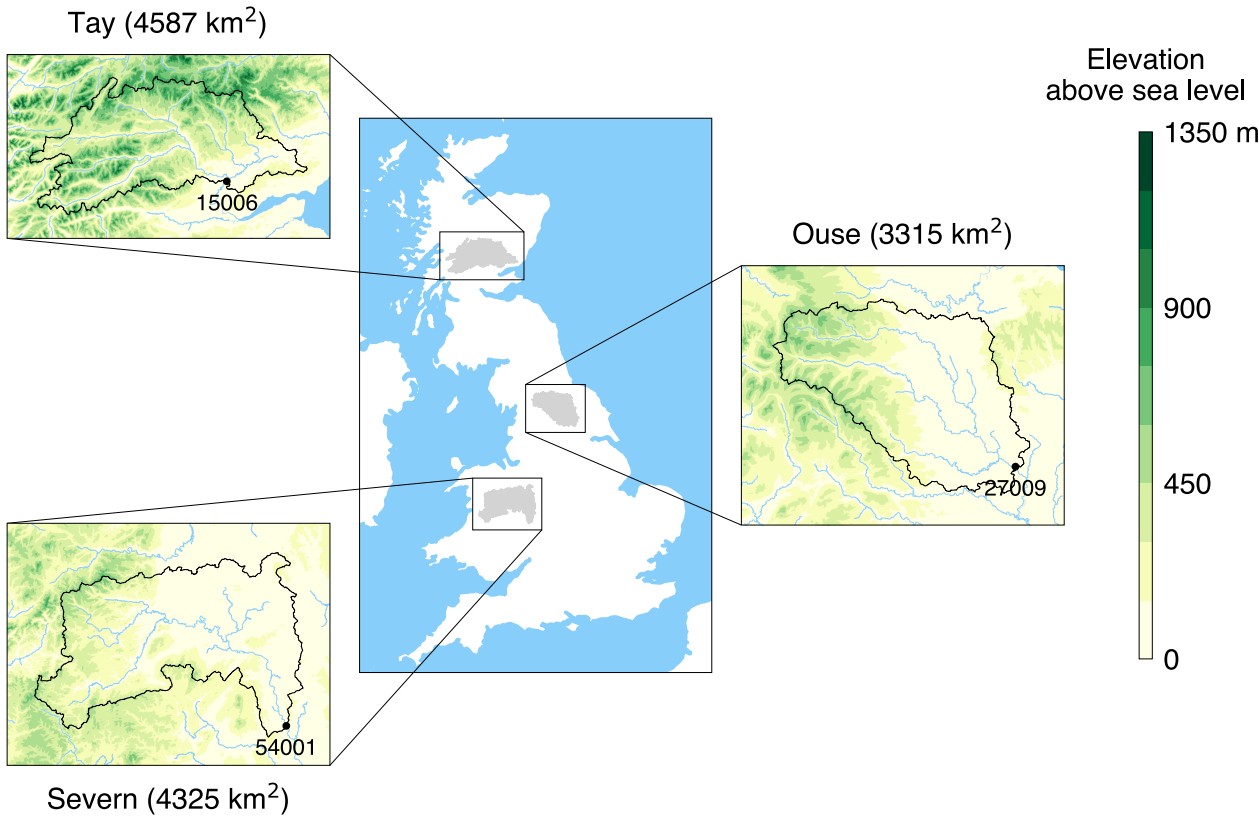

**Figure 3.** Location of the three study catchments in Great Britain. In the zoomed-in panels, the dots correspond to the outlets of the catchments and their adjoining five-digit labels correspond to the number of the National River Flow Archive (NRFA) hydrometric stations at these outlets. The elevation is based on digital spatial data licensed from the UK Centre for Ecology & Hydrology, ©UKCEH (Morris and Flavin, 1990, 1994).

framework simulations. Indeed, the overlaid hydrographs suggest that the overall observed discharge pattern is captured by the simulations, while the spatial distributions of river discharge sketch a realistic picture of the catchment river network.

In addition, a quantitative evaluation of the performance of the framework simulations is done with respect to the river dis-
charge at the catchment outlet where observed and simulated time series are compared using the non-parametric Kling-Gupta efficiency ($R_{NP}$). This is a composite metric made of three equally-weighted components $r_S$, $\alpha_{NP}$, and $\beta$, assessing the agreement in the dynamics (i.e. correlation), the variability, and the volume (i.e. bias) of the discharge time series, respectively (Pool et al., 2018). Table 3 features these metric components computed for the three configurations and the three study catchments using the Python package `hydroeval` (Hallouin, 2021).





**Table 3.** Quantitative comparison of the three configurations for the three study catchments.

| Catchment | Configuration | Performance metrics | | | |
|---|---|---|---|---|---|
| | | $R_{\mathrm{NP}}$ | $r_S$ | $\alpha_{\mathrm{NP}}$ | $\beta$ |
| Tay | 1 | 0.667 | 0.692 | 0.981 | 1.126 |
| | 2 | 0.743 | 0.773 | 0.979 | 0.881 |
| | 3 | 0.766 | 0.821 | 0.940 | 1.139 |
| Ouse | 1 | 0.456 | 0.614 | 0.870 | 1.361 |
| | 2 | 0.674 | 0.705 | 0.945 | 0.875 |
| | 3 | 0.493 | 0.790 | 0.882 | 1.447 |
| Severn | 1 | 0.495 | 0.721 | 0.909 | 1.411 |
| | 2 | 0.706 | 0.719 | 0.946 | 0.933 |
| | 3 | 0.218 | 0.840 | 0.926 | 1.762 |

The comparative performance of the three configurations for each catchment in turn informs the most suitable combination of components for a given temporal and geographical context. For instance, the third configuration appears to be the most suitable in the Tay catchment, if one is solely interested in simulating the river discharge accurately ($R_{NP}$ of 0.766), while the second configuration would be preferred for the Ouse and Severn catchments ($R_{NP}$ of 0.674 and 0.706, respectively). However, these conclusions are metric-dependent and the analysis of the components of the composite metric can reveal the strengths and weaknesses of a given configuration, e.g. while the third configuration performs highest on the composite metric in the Tay catchment, its ranking on capturing the flow variability is the lowest of the three configurations ($\alpha_{NP}$ of 0.940).

Some caveats in this comparison are that the third configuration used a calibrated model unlike the first and second configurations, and the second configuration used data from a model constrained to conserve mass and energy, unlike the other configurations that only conserve mass. This likely skews the comparison.

This brief analysis of the results is used to demonstrate the potential of the framework to elicit the most suitable combination of components to simulate the hydrological behaviour of a given region; it is not to draw definitive conclusions as to which combinations should be used for the catchments selected here. Moreover, this analysis focuses on one hydrological variable, the river discharge, but other hydrological variables such as e.g. soil moisture or evaporation could also be considered.

## 6 Future developments

### 6.1 Implicit spatial heterogeneity

The spatial heterogeneity of real-world hydrological processes can be accounted for in models by either adjusting the resolution of their spatial discretisation explicitly (i.e. through finer mesh elements) or implicitly (i.e. through sub-mesh units). The former is currently possible using the framework but can be computationally costly, although flexible discretisation allows





**Figure 4.** Simulation of the river discharge with the three configurations for the Tay catchment: **(a)** observed and simulated hydrographs with configuration 1 at the catchment outlet [3.39 °W, 56.51 °N], **(b)** gridded simulated discharge with configuration 1 for the last simulation step [2018-01-01], **(c)** observed and simulated hydrographs with configuration 2 at the catchment outlet [314, 736], **(d)** gridded simulated discharge with configuration 2 for the last simulation step [2018-01-01], **(e)** observed and simulated hydrographs with configuration 3 at the catchment outlet.





most of the cost to be focussed on the part(s) of the water cycle where finer resolutions are justified. Implicit resolutions are
not currently supported in the framework but are often used by land surface models, e.g. tiling or mosaic schemes to consider
spatial heterogeneity within the grid cells of the atmospheric models to which they are coupled. These schemes consider varied
biophysical behaviour depending on e.g. land cover types. Some hydrological models also use such schemes in the form of
hydrological response units (HRUs), where the unit (i.e. the tile) is typically based on a combination of soil types and/or ele-
vation bands and/or land cover types. These sub-mesh units need not be geographically located, but they can be hydrologically
connected nonetheless (e.g. flow matrix in TOPMODEL (Beven and Freer, 2001) in use e.g. in HydroBlocks (Chaney et al.,
2016), intra-hillslope configuration in CLM (Swenson et al., 2019), unit-to-unit routing in ORCHIDEE (Nguyen-Quang et al.,
2018)). Such schemes will be valuable for later versions of the framework, although it is anticipated that such implementation
will not be trivial. In addition, the explicit supermesh required in the remapping operation (see subsection 2.3) will need to be
extended to consider this implicit discretisation (i.e. sub-mesh) as well.

## 6.2 Unstructured spatial meshes

Structured grids may not be sufficient for some components. For example, existing land surface layer models often inherit
their spatial discretisation from an associated atmospheric model. In atmospheric models, the Earth's poles cause problems
when using regular latitude-longitude discretisations because the grid cells get smaller as the coordinates converge towards
the poles, therefore unstructured meshes are often preferred instead, e.g. reduced grids (e.g. ECMWF's IFS model (Hortal and
Simmons, 1991)), icosahedral grids (e.g. DWD's ICON model (Zängl et al., 2015), IPSL's DYNAMICO core (Dubos et al.,
2015)), or cubed spheres (e.g. UK Met Office's LFRic model (Adams et al., 2019), NOAA's FV3 model (Putman and Lin,
2007)). Moreover, triangular grids are also used in some hydrological and hydraulic models (e.g. PIHM (Li and Duffy, 2012),
CHILD model (Tucker et al., 2001)). Finally, rainfall-runoff models are typically applied on drainage basins, where space
domains based on highly complex polygons may also prove useful to support such models as components for the framework.
Supporting such spatial discretisations would extend the range of models that can be refactored into components for the
framework, however, current remapping functionalities may fall short to deal with these, since e.g. ESMF currently does not
support unstructured grids with cells with more than four edges. In addition, unlike structured grids, unstructured grids require
additional horizontal connectivity information.

## 6.3 Parallel execution

The framework can leverage high performance computing through internal parallelisation and through the generation of many
model instances to form an ensemble of water cycle simulations.

For internal parallelisation, the modular framework is readily structured to offer opportunities for both task decomposition
(i.e. execute components concurrently) and domain decomposition (i.e. compute spatial subsets of the modelled region con-
currently). While we anticipate most applications using an implementation of the Message Passing Interface (MPI), there is
no a priori reason why other parallelism techniques such as OpenMP or OpenACC could not be used. However, if different
approaches are used in different coupled components, load balancing would likely be an issue. If task decomposition is being



used, the relationship between components will need to consider the spatio-temporal scales being modelled in each component, e.g. coupling of "fast" physics components may require adjustments rather than increments, ruling out concurrent in favour of sequential coupling (Balaji et al., 2016). If domain decomposition is used, the distribution of the spatial supermesh on the processing elements will need to be communicated to the framework to enable the transfer of information across component, albeit the domain decomposition itself is anticipated to be the responsibility of the component contributors.

Usage in large ensemble experiments represent an "embarrassingly parallel problem" with many available implementation mechanisms. The flexibility offered by the object oriented framework design makes it easier to perturb selected aspects of a given framework configuration to produce many alternative models.

## 6.4 External model coupling

Models of the land system are typically used with atmospheric and ocean models, e.g. in Earth system models. This means that the framework will not only need to be usable standalone but also in a coupled configuration. This will involve considerations regarding memory allocation, time synchronisation, and domain decomposition. And this will require the formulation standard interfaces between the framework components and the external models (e.g. Polcher et al., 1998; Best et al., 2004). In addition, such a modular framework for the land system can contribute to the improved coarse-grained concurrency of Earth system models (Balaji et al., 2016; Lawrence et al., 2018).

## 7 Conclusions

This framework represents the first implementation of a new modular blueprint to model the terrestrial water cycle. It is open source and comes with extended online documentation. By design, this Python package is intended to be easy to use, with a low entry bar for people with little programming experience. Indeed, installing a Python package is straightforward and only a few steps in a Python script are needed to set up and run a complete model in a Jupyter notebook, which is likely to prove useful for teaching and training activities alike. It is also intended to be easily customisable, through choosing from a library of compatible components those most suitable for a given region and for a given objective. Finally, it is intended to be easily extensible by creating new components, which should streamline the development and sharing of new science for the terrestrial water cycle.

Later versions of the framework will implement the additional developments discussed in section 6. In addition, the extension of the blueprint to include other geochemical cycles (e.g. carbon, nitrogen, phosphorus) as well as anthropogenic influences is planned. In the meantime, we hope that the science library will grow with new contributions from the land, hydrology, and groundwater modelling communities, and stimulate collaborations between them.





*Code availability.* The framework is open source and available on GitHub in the repository unifhy-org/unifhy. The science components Artemis, RFM, and SMART are also open source and available on GitHub in the repositories unifhy-org/unifhycontrib-artemis, unifhy-org/unifhycontrib-rfm, and unifhy-org/unifhycontrib-smart, respectively. The online documentation is accessible at unifhy-org.github.io/unifhy

*Data availability.* The input data used in the case studies is publicly available using the references provided in Appendix A. The observed river flow data is publicly available from NRFA (http://nrfa.ceh.ac.uk/, last access: 10 October 2021). The framework output data is available
upon request from the corresponding author.

## Appendix A: Data sources

**Table A1.** Sources for data used in configuration 1.

| Variable name | Dataset name | Data DOI | Related publications |
| --- | --- | --- | --- |
| precipitation flux | ERA5 | 10.24381/cds.20d54e34 | Cucchi et al. (2020) |
| specific humidity | | | |
| surface downwelling shortwave flux in air | | | |
| surface downwelling longwave flux in air | | | |
| air temperature | | | |
| wind speed | | | |
| surface albedo[1] | – | – | – |
| vegetation height | GLAS | – | Los et al. (2012) |
| leaf area index | MOD15A2H | 10.5067/MODIS/ MOD15A2H.006 | – |
| topmodel saturation capacity | HWSD | – | Nachtergaele et al. (2012) |
| saturated hydraulic conductivity | | | |
| topographic index | – | – | Marthews et al. (2015) |
| flow direction | HydroSHEDS | – | Lehner et al. (2008); |
| flow accumulation | | | Lehner and Grill (2013) |

[1] Produced using suite u-ag343 accessible at https://code.metoffice.gov.uk/trac/roses-u.



**Table A2.** Sources for data used in configuration 2.

| Variable name | Dataset name | Data DOI | Related publications |
|---|---|---|---|
| surface_runoff_flux<br>subsurface_runoff_flux | CHESS-land | 10.5285/c76096d6-45d4-4a69-a310-4c67f8dcf096 | Martínez-de la Torre et al. (2018) |
| flow_direction<br>flow_accumulation | – | – | Davies and Bell (2009); Martínez-de la Torre et al. (2019) |

**Table A3.** Sources for data used in configuration 3.

| Variable name | Dataset name | Data DOI | Related publications |
|---|---|---|---|
| precipitation flux<br>specific humidity<br>surface downwelling shortwave flux in air<br>surface downwelling longwave flux in air<br>air temperature<br>wind speed | ERA5 | 10.24381/cds.20d54e34 | Cucchi et al. (2020) |
| surface albedo[1] | – | – | – |
| vegetation height | GLAS | – | Los et al. (2012) |
| leaf area index | MOD15A2H | 10.5067/MODIS/MOD15A2H.006 | – |

[1] Produced using suite u-ag343 accessible at https://code.metoffice.gov.uk/trac/roses-u.

## Appendix B: Additional results

*Author contributions.* All co-authors designed the blueprint for the framework. TH and RJE developed the framework implementation in Python. TH performed the simulations. TH and RJE processed the simulation outputs. TH prepared the manuscript with contributions from all co-authors.




**Figure B1.** Simulation of the river discharge with the three configurations for the Ouse catchment: **(a)** observed and simulated hydrographs with configuration 1 at the catchment outlet [1.13 °W, 53.99 °N], **(b)** gridded simulated discharge with configuration 1 for the last simulation step [2018-01-01], **(c)** observed and simulated hydrographs with configuration 2 at the catchment outlet [456, 455], **(d)** gridded simulated discharge with configuration 2 for the last simulation step [2018-01-01], **(e)** observed and simulated hydrographs with configuration 3 at the catchment outlet.



**Figure B2.** Simulation of the river discharge with the three configurations for the Severn catchment: **(a)** observed and simulated hydrographs with configuration 1 at the catchment outlet [2.32 °W, 52.38 °N], **(b)** gridded simulated discharge with configuration 1 for the last simulation step [2018-01-01], **(c)** observed and simulated hydrographs with configuration 2 at the catchment outlet [378, 275], **(d)** gridded simulated discharge with configuration 2 for the last simulation step [2018-01-01], **(e)** observed and simulated hydrographs with configuration 3 at the catchment outlet.



*Competing interests.* The authors declare that they have no conflict of interests.

*Acknowledgements.* This work was funded by the Natural Environment Research Council (NERC) as part of the Hydro-JULES programme (NE/S017380/1). The authors would like to thank the developers of `cf-python`, David Hassell and Sadie Bartholomew, for implementing the requested features required by the framework, and Andy Heaps for the `cf-plot` support. They would also like to thank Katie Facer-
Childs for providing the rainfall and potential evapotranspiration data used to calibrate the SMART model, Helen Davies for providing the HydroSHEDS dataset, and Huw Lewis for providing a Python version of RFM used to create an open water component for use in the framework.



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
