# Peer review of "UniFHy v0.1.1: A community modelling framework for the terrestrial water cycle in Python"

_Geoscientific Model Development, 2021_

## Author Comment (AC1)

Dear Editor, Dear Referees,

We would like to thank you for your consideration of our manuscript and for the thorough review of the manuscript and testing of the assets, and the positive and constructive feedback you provided us with. Please find below a detailed point-by-point answer to the referees' comments.

The main revisions include:

- Highlighting the similarities and differences with existing modelling framework (notably LandLab)
- Emphasising the reasons for an intermediate modularity in the design of the framework
- Improving the online documentation (installation instructions, improved tutorial now with accompanying data) and its access (link added in the manuscript and in the README file in the codebase)

A new version of UniFHy has been released (v0.1.1) to include these revisions alongside some bug fixes.

Thibault Hallouin, on behalf of the co-authors.

**Nomenclature:**

RXCY – Referee number X Comment number Y AR – Authors' Reply PXLY – Page X Line Y

**Referee #1 (Anonymous)**

**R1C1:** I have one major concern. As a researcher familiar with Landlab, I find that unifhy is very similar to Landlab, from concept to code. Because of this, I feel that the paper does not currently meet items 2 and 3 from the GMD review criteria:

- Does the paper present novel concepts, ideas, tools, or data?
- Does the paper represent a sufficiently substantial advance in modelling science?

What I would like from the authors is a justification for how unifhy differs from Landlab, what new ideas are incorporated into its design and UX, and how unifhy represents an advance over or a novel alternate to Landlab in modeling. I don't think this demand is onerous—an extra paragraph or so should suffice. I'm imagining the perspective of a new graduate student in hydrology reading this paper, and I'd like them to understand how unifhy builds on the previous contributions of others.

I enjoyed reading this paper, and I want it to be published, but the authors haven't quite shown that unifhy is a unique contribution to science.

**AR:** We agree with the reviewers that the version of UniFHy presented and LandLab have many functionalities in common. However, we believe that the motivations behind the development of UniFHy differ from those of LandLab, and UniFHy is by no means meant as a replacement for LandLab which already fulfils particular needs very well. The version of UniFHy presented in the manuscript (i.e. v0.1.0) is a first milestone on a roadmap that will be described in a separate manuscript currently in preparation (a roadmap outline can already be found in Dadson et al., (2021)).

In a nutshell, the objectives on our roadmap are ultimately for UniFHy to be able to provide replacement models for existing land surface models in the form of modular land system models that can be coupled to atmospheric models and in Earth system models:

- To be coupled to such external models, they need to be able to hand over the time advancement and the memory management to the external model. These are not readily possible with LandLab: in particular time advancement is the responsibility of the user unlike in UniFHy where a Clock handles it. While we appreciate that some of the functionalities of LandLab could have been reused in UniFHy (e.g. the component-based approach, and the spatial discretisation), a complete engineering control for the evolution of the framework was deemed essential.
- To improve existing land surface models, they need to make the collaboration between the different modelling communities (land surface modellers, catchment modellers, groundwater modellers, etc.) the norm rather than the exception. To achieve this, UniFHy promotes a modular approach to land system modelling, where each component maps onto one or more communities that can bring their model(s) along to contribute to the framework. In order to guarantee the compatibility and reusability of the components over the lifetime of the framework, UniFHy proposes a standardised interface between the land components as a prerequisite for inclusion in the library of compatible components. LandLab offers more degrees of freedom to the user, whose components can expose any interface they like, which has its advantages, but which is not in line with our objectives.

We understand that LandLab is focussed on the challenges faced in geomorphology while UniFHy is focussed on the challenges faced in hydrology, meteorology and climate science. This explains and justifies why their requirements and hence their roadmaps are different, and why one cannot replace the other.

The "Introduction" has been revised to better highlight the challenges being addressed or on course to being addressed by UniFHy, and a paragraph mirroring the challenges highlighted in the introduction has been added in the "Conclusions" section and now positions UniFHy in relation to the other modelling frameworks (see P21L373-381 in the revised manuscript) in the hope that this will help readers understand the differences and decide which approach is more appropriate for their needs.

**R1C2:** Line 30: For Landlab, please also cite Barnhart et al. 2020: https://doi.org/10.5194/esurf-2020-12 (this is a PDF link)

**AR:** Done. See P2L32 in the revised manuscript.

R1C3: Line 31: Basic Model Interface instead of Basic Modelling Interface

**AR:** Done. See P2L33 in the revised manuscript.

**Referee #2 (Ethan Coon)**

**What is new in this framework relative to existing work?**

**R2C1:** The authors provide a reasonably complete description of existing coupling tools, including key players such as underlying technologies for "coarse-grained" coupling technologies (MCT, CPL, ESMF, CSDMS/LandLab, etc). However, they fail to describe what makes their framework new or even an improvement upon these technologies. In fact, I'm fairly sure that everything described in this framework could have been directly implemented in the CSDMS codebase with no new framework. I believe CSDMS already implements all described features of this codebase, including existing time-stepping and spatial schemes, python-based drivers and configuration of models, remapping capabilities through third-party libraries, and existing interfaces for passing data between these models. If one were to eliminate the requirement of "in python," I suspect several other tools would meet the needs of this effort as well (e.g. MCT, BMI, others), although maybe not with bent toward hydrology.

**A major revision is required to make clear what, if anything, is unique or novel about this framework relative to these other coupling frameworks.**

To be clear, if there is nothing novel about this work other than it is more specifically tailored for hydrologic models and some hydrologic models have already been adapted to this framework, I don't think that it would immediately disqualify this from being published as a code description. But it must be clear how the work relates to existing work.

**AR:** addressed together with **R1C1**, please refer to our reply to **R1C1**.

**Reproducibility and Software Quality**

**R2C2:** As noted above, this submission clearly fits GMD's concept of a "model description" paper. In particular, "model description" papers should be held to a high standard of tracibility and/or reproducibility – the reader should be able to download and install the code, run a basic example, and gain some concept of how to use the model for any "model description"

paper. The following describes my experience in trying to download the code, run the tests, use the tutorial, and finally to use the user's guide and training material.

First, I followed the cited DOI, which took me to the github repository. At first glance, the software repository seemed well organized. Browsing through the repository suggested that the code could be installed via pip (a requirements.txt file existed), and the code source was readable, reasonably well annotated/commented, and included template for new components. Unfortunately, it was not obvious to me as a user how to begin to use the codebase. Please add a link to documentation on the README page and in the manuscript – as the DOI points to the github repo, the "new" user will not know how to find the documentation. In the end, I was able to guess where to find the documentation by looking for github pages, but this should not be required.

**AR:** We have added a link to the documentation at the end of the section "Usage of the framework" (see P2L212 in the revised manuscript) and in the README file in the codebase.

**R2C3:** Next, I tried to install the code, following the installation documentation. I cloned the latest version, noting the requirements.txt file, and tried to install via pip, with both the recommended `pip install unifhy` and `pip install -r requirements.txt`. These failed with the message:

>ERROR: Could not find a version that satisfies the requirement esmpy (from unifhy) (from versions: none)
 >ERROR: No matching distribution found for esmpy

I was able to install the base framework package via conda – success!

**AR:** Thank you for highlighting this. Indeed, **esmpy** is not available on PyPI, which explains why **pip** fails at installing it – it needs to be installed separately before. We have mentioned this complication in the "Installation" webpage in the online documentation, and recommended **conda** as the preferred way for installing **unifhy**.

**R2C4:** Next, I noted that the codebase had unit tests, which I was pleased to see. Running the tests via `cd tests; python run\_all\_tests.py` resulted in:

```
>Ran 90 tests in 539.397s
>FAILED (errors=104, expected failures=5)
```

Note that this is probably not a big deal, as probably I am running the tests incorrectly, but it would be good if the user's guide described how to run the tests. Successful tests give the user confidence that the code is still supported and that they have correctly installed the needed requirements.

**AR:** For information, the test suite *run\_all\_tests.py* requires the compilation of Fortran and C framework components using the *Makefile* provided in the repository under *tests/components* – it is intended for continuous integration purposes. Nevertheless, we agree with the reviewer that running a test suite is a useful way for a user to make sure they have installed the package correctly. To this end, we have added a section "Testing the framework" in the "Installation" webpage in the online documentation, where we recommend the user to run the test suite *run\_basic\_tests.py* (which does not rely on compiled extensions, while still exhaustively testing the proper functioning of the framework).

**R2C5:** I was pleased to see a tutorial; I tried to follow the tutorial in the documentation at: https://unifhy-org.github.io/unifhy/users/tutorial.html. It does not appear that this was intended to actually be executed? The required netCDF files were not in my repository, and there did not seem to be a repository called "tutorial" or similar which contained these files. None of the documentation described where to get these files, so I gave up and moved on.

**AR:** Thank you for the suggestion. It now appears clear that some users may want to reproduce the tutorial themselves, so we have added the input data in a newly created **data** folder in the codebase and modified the file paths in the tutorial accordingly.

**R2C6:** Next, I saw there was a training repository in unifhy-org. I cloned this, started up a jupyter notebook, and attempted to run the demo problem there. Immediately I ran into the problem that the contributed models (e.g. artemis, rfm) are not provided with the main repository. Furthermore, there was no documentation in the manuscript, training repository, or the corresponding artemis/rfm repositories on how to install these component models. Again, I began to guess, trying conda (no luck) before successfully installing them by cloning their respective repositories and running `python setup.py install`.

**AR:** Indeed, the framework components are intentionally kept separate from the framework itself. For information, the components currently compatible with the framework are all available on PyPI (i.e. installable using **pip**). A new section "Installing framework components" has been added in the "Installation" webpage in the online documentation to highlight this, and

each individual component now has a "How to install" section in their **README.rst** file to provide the installation procedure. Finally, it is also been made more explicit by the addition of a note in the "Configuration" section (see P9L186-187 in the revised manuscript).

**R2C7:** This got me to the next step in the demo problem, before, once again, running into an issue where the demos relied on netCDF files that did not exist anywhere that I could find – the notebook suggests they should be at `/data/demo-data/` but I suspect this demo is intended for local users of a private machine somewhere.

**AR:** We have added the input data in a newly created *data* folder in the codebase and modified the file paths in the notebooks accordingly.

**R2C8:** In all, while tutorials and demos existed (good) I was unable to run even a simple example (bad). The documentation, while it does exist (good) is incomplete and misses key "how to" descriptions to help the user get started (bad). If a GMD "model description" paper is intended to make clear that a new code is available and usable by the community, that code should be clearly usable by the community, which it does not appear to be. Prior to acceptance, this would have to be significantly improved to the point that a user could follow reasonable descriptions to run a demo problem using the code on their own machine.

**AR:** In order to provide the user with easily accessed "how to" descriptions, we have added a "Getting started" section on the homepage of the documentation containing quick links to the installation instructions and the tutorial. Together with the other modifications for **R2C2**, **R2C3**, **R2C5**, and **R2C6**, we hope that these meet the significant improvements expected by the reviewer.

**R2C9:** Finally, as far as I can find, no "assets" or data repositories are provided with regards to input files or other files needed to trace or reproduce the "selected configuration" simulations run and described in the manuscript.

In summary, I would expect to, with reasonable effort, be able to:

- 1. Download and install the code and all requirements
- 2. Run the tests
- 3. Run an example or tutorial
- 4. Have access to the complete input files needed to reproduce the demos shown in the paper.

I was only able to do #1 successfully, and that required some guessing to even find the documentation.

As a result of this effort, I believe that the reproducibility and traceability of this paper is insufficient to meet the standards of GMD.

**AR:** In our replies to previous comments, we have already addressed 1., 2. and 3.:

- A link to the documentation was added in the *README.rst* file (R2C2), and the installation instructions in the documentation were updated to recommend **conda** to install the framework and explain how to install the science components (R2C3, R2C6, R2C8).
- 2. Explanations on how to run a test suite to check for the proper installation and functioning of the framework was added in the documentation (**R2C4**)
- 3. Data to run the tutorial in the documentation was added to the codebase so that a user can now follow it along (R2C5)

However, to achieve 4., this requires sharing larger datasets we have not produced and that are already freely accessible elsewhere (as listed in appendix A), so that we believe it is sufficient for reproducibility and traceability purposes. Moreover, it is important to stress that the scientific results presented in the case studies are not used to draw scientific conclusions, but rather to demonstrate the potential of the framework to draw scientific conclusions in the future.

**How flexible is the defined API?**

**R2C10:** In this work, the authors define a very narrow API for a fixed set of fluxes between exactly three components. This raises several concerns. As examples:

- What happens if a component does not supply all the needed variables? For instance, one might want to use a simple empirical model, e.g. the Priestley-Taylor model, to provide a total evapotranspiration flux. While this could probably be used within another component to supply the remaining fluxes (e.g. PRMS implements a Priestley-Taylor model and could supply the rest of the fluxes), it does not appear that the API is flexible enough to "split" the API across multiple, smaller-grained components.
- What happens if a model would like to integrate more tightly than the current flux-based coupling? For instance, integrated hydrologic codes (e.g. those described in Kollet et al WRR 2017) often tightly couple surface and subsurface flow, meaning that there is no "permanent open water." Codes such as these would benefit from an API to provide a coupling to land surface codes. It is unclear whether this API would support

extension to include not just fluxes but also primary variables e.g. head or pressure (likely not) or whether lumping the surface and subsurface components together into a single component is supported (maybe?).

In general, this coupling capability seems to be very narrowly aimed at a specific class of codes (e.g. Artemis, RFM, and SMART and others that are structured grid, routing + infiltration + land surface model components), and that this "fixed granularity" of the coupling API means it is not very useful outside of a very narrow class of codes. While this is not a deal-breaker, it should be addressed by a clear statement of the class of codes that fit this API, and how or if the granularity of this API could be changed or made more flexible.

**I think this addresses an important class of models, but it should be clear what class of models this is.**

For instance, another approach would be to allow arbitrary collections of models to be registered into a shared state; at runtime the driver ensures that all fluxes that are needed are supplied by a component, thereby checking that the model is "well-posed." This allows the granularity to change as needed, even within a flux-coupling concept such as this (e.g. CSDMS, Tucker et al GMD 2022). Other more general multi-physics codes (e.g. Coon et al Env. Model Software 2016) use dependency graphs to ensure that coupling needs are met and allow arbitrary components that meet a generic API to be coupled, allowing even greater flexibility in what it means to be a component. These seem much more flexible than this API, which is more of a fixed-granularity, "role-based" component concept.

**The manuscript should do a better job to clarify what it means by flexible.**

**AR:** The considerations on flexibility raised by the reviewer in this comment perfectly highlights the differences that exist between LandLab and UniFHy, and why UniFHy is not intended to replace LandLab in any way. Indeed, UniFHy intentionally reduces the degrees of freedom by enforcing a standardised interface between its components. We believe that, while allowing any arbitrary collection of models to be coupled together has its own merits, it does not help in structuring the discussion between communities with their existing models and practices. If next generation land system models are to embed the best available knowledge on the functioning of the terrestrial water cycle, there needs to be a suggested blueprint to be followed, so that components can be replaced with new/different ones, and so that components remain compatible with one another over time. While we hope that the proposed standardised interface is up to the task, we acknowledge that it may require some refinement

required once these communities seize the framework and start contributing to it. This does not preclude the users to assemble smaller-grained components together to form UniFHy standardised components and, as such, LandLab would be very useful to do so.

We hope that the existing explanation in the "Modular water cycle blueprint" section with one addition (see P3L86-87 in the revised manuscript), and the amendments made in reply to **R1C1** make it clearer that we opted for an intermediate level of modularity.

Concerning the class of models that can be included in the framework, we don't see any limitation beyond the fact that they need to comply with the standardised interface. It is true that the models currently used in the case studies belong to the "simpler" end of the spectrum of models, but more complex land surface models such as JULES are currently being refactored to comply with the standardised interface (see e.g. Blyth et al. (2021) for a review of the class of models targeted). We anticipate situations where it may not make sense to couple simpler with more complex models, but we believe that this remains the responsibility of the user to decide.

We clarified this point in the "Contribution to the framework" section (see P13L234-236 in the revised manuscript).

**Minor points**

**R2C11:** The authors state that: "These models do not allow different parts of the land system to be simulated at different explicit resolutions." The same criticism is true of this effort; only structured Lat/Long meshes are described here.

**AR:** While it is true that, in its current version, the different parts of the land system (i.e. the framework components) need to be spatially discretised in the same coordinate system (i.e. latitude-longitude grid, rotated latitude-longitude grid, or British national grid), UniFHy does allow for each component to be spatially discretised with a different resolution (e.g. 0.5 degree vs. 0.1 degree vs 0.05 degree, or 1 km vs. 5 km vs. 25 km). This already represents some progress in flexibility compared to most existing land surface models.

**R2C12:** The introduction is missing a short discussion of existing ESM/Integrated Hydro Terrestrial Modeling approaches that bridge this scale in hydrologic modeling. For instance, work in so-called hyper-resolution hydrology has started to bridge this scale/community divide, and examples such as the US's National Water Model via WRF-Hydro, Maxwell et al GMD 2015's ParFlow papers, and several others have started to bridge this gap (see, for a summary, Paniconi and Putti, WRR 2015 for a good, if not very current, summary). While these are not general-purpose coupling frameworks, they are currently providing models that cover the same set of processes and scales targeted by this framework.

**AR:** We agree with the reviewer that these models represent very valuable contributions to the land system modelling efforts, however, as highlighted by the reviewer that these models are not frameworks, we believe that they are not directly relevant in the literature background provided in the introduction, and this is the reason why these and other modelling efforts have not been mentioned in the introduction. Further, these examples (e.g. US National Water model - WRF combination) are instances of coupled models and do not necessarily represent examples of a framework with the flexibility to couple different model codes. That being said, we hope that, provided that these models can be refactored, UniFHy can support scientifically interesting comparative exercises to explore the merits of these models against/in combination with other existing land surface models.

**R2C13:** In the section on discretization, the authors describe that timestepping schemes must be integer multiples of each other in order to synchronize at the needed intervals. It is unclear why this would be true – it seems that a component model ought to be able to do whatever it needs to integrate across the interval. For instance, this would preclude adaptive timestepping, which may be needed in implicit subsurface codes.

**AR:** Because the framework components aim to be coupled together, there is the necessity to have synchronisation times between them to allow for fields to be exchanged without unintended numerical diffusion. Together, these time steps form the temporal resolution for a given component, but what happens within those time steps is not constrained by the framework, and sub time steps or adaptive timestepping schemes would be perfectly acceptable. As far as UniFHy is concerned, the component temporal resolution represents a commitment that the component will provide the fields required by the standardised interface at regular intervals which are determined a priori and which are compatible with the other components' intervals. Clarifications regarding this point were added in the section "Flexible discretisation" (see P6L125-127 in the revised manuscript).

**R2C14:** The authors motivate the need for a shared naming and units convention for codes to be coupled successfully. However, there has been significant efforts to identify ontologies that formalize variable names and units for use in coupling codes; this work was not considered or cited in this manuscript. Specifically, CSDMS's Standard Names and the work by Scott Peckham

and collaborators formalize an ontology for land surface modeling, and the CF conventions standard names provides a framework for defining variable names in climate applications. It would be preferable to adopt one of these standards where possible rather than create a new one for this work. As one example, CF conventions defines "canopy\_throughfall\_flux" that could be adopted instead of "canopy\_liquid\_throughfall," etc.

**AR:** As stated in the section "Meaningful data", UniFHy relies on the CF conventions to read and write data featuring standardised metadata because of its future trajectory into Earth system modelling where the CF conventions are the standard. By extension, it is true that the interface between components is also intended to be compliant with the CF conventions, including the use of its ontology (i.e. the CF standard names list). However, the existing names and definitions are mostly focussed on atmospheric fields, and to some extent oceanic and land fields, but nothing specifically tailored towards hydrology. The Hydro-JULES project, which leads the development of UniFHy, intends to submit a list of fields to be added to the CF standard names list, and if the relevant concepts already exist from the CSDMS standard names list, they will be obvious candidates for CF.

As for the example provided by the reviewer, the full name for the field is "canopy\_liquid\_throughfall\_and\_snow\_melt\_flux": unlike CF's "canopy\_throughfall\_flux", it deliberately only considers the liquid fraction of precipitation reaching the ground (which is the one transferred to the subsurface component), while the solid fraction of precipitation first contributes to the snow pack (belonging to the surface layer component), before potentially melting. This is one example where the CF standard names list falls short to consider fields relevant to hydrology.

A note was added in the section "Meaningful data" to indicate that the CF ontology is followed where possible, but that some divergence from it exists in the framework interface (see P7L161-163 in the revised manuscript).

**R2C15:** There are a lot of potentially useful things described in the future work discussion, but none of these are currently in the codebase, and many of them are available in other comparable products. For instance, unstructured meshes may be coupled through a variety of general-purpose remapping tools (MOAB, DTK, TempestRemap, and many others). Parallel execution is supported in MCT, CCA, and even ESMF which is used here, and has been developed for BMI by the National Water Model. I do agree that a formal API enabling subgrid connections (e.g. PFT subgrid concepts, etc) for heterogeneity is an exciting research area,

however. These sections should either acknowledge that such work already exists or be removed completely.

**AR:** We have added the relevant existing libraries relevant to achieve the objective presented in the Future work discussion as suggested by the reviewer (see P20L336-338 and P20L351-353 in the revised manuscript).

**R2C16:** Lastly, object-oriented design has a lot of advantages for defining interfaces, but often sets limitations on data structures and data dependencies. I would have liked to have seen a description of the limitations of this data model. For instance, can the exchanged fluxes be GPU memory? MPI-parallel vectors?

**AR:** Adding a meaningful discussion of these issues would add another level of complexity to the paper, not least because we have not yet explored them in any detail – we expect to do that when we try and actively integrate this framework into an ESM. As discussed above, one of the points of distinction between this framework and others is that we are expressly preparing for "handing over some memory management to an external customer" as opposed to using a pure coupling interface always using internal memory. In practice, when we get that going we expect to be providing another layer of indirection between our interface and the actual memory layout (across devices, sockets and nodes). The bottom line is that while we agree that interfaces constrain (all interfaces, object-oriented or otherwise), we expect that if we need expedience for optimisation, we will need to deliver that with another layer of indirection. How all that interacts with the internal parallelisation of the UniFHy components may reveal not being a trivial task – and so we think discussing this now would be premature.

**References:**

Dadson, S., Blyth, E., Clark, D., Ellis, R., Fry, M., Polcher, J., Hallouin, T., Lawrence, B., Lister, G, Hughes, A, Best, M. *Hydro-JULES system design*. Wallingford, UK, UK Centre for Ecology & Hydrology, 33pp. Unpublished (2021). http://nora.nerc.ac.uk/id/eprint/530671

Blyth, E.M., Arora, V.K., Clark, D.B. et al. *Advances in Land Surface Modelling*. Curr Clim Change Rep 7, 45–71 (2021). https://doi.org/10.1007/s40641-021-00171-5

---

## Author Response (AR2)

Dear Editor, Dear Referee,

We would like to thank you for further considering our manuscript and for the comments and suggestions made on our revised manuscript.

We believe that most of the concerns and comments from the reviewer are due to shortcomings in our explanation of the context and objectives of the modelling framework we describe - which aims to provide a blueprint and infrastructure for the future of the land components featured in Earth system models, so that they are more modular and more suited to simulate the terrestrial water cycle.

This is why the main focus in the revised version of our manuscript has been to clarify these objectives, to detail the limitations in the current land components of Earth system models, and to explain and describe how the framework is addressing those limitations. In addition, we have replied and addressed, where appropriate specific reviewer comments as detailed below.

We hope that our revisions address any misunderstanding and meet the expectations of the Editor and the Referee.

Thibault Hallouin, on behalf of the co-authors.

**Nomenclature:**

RXCY – Referee number X Comment number Y
AR – Authors' Reply
PXLY – Page X Line Y

**Referee #3**

**General issues**

**R3C1:** The manuscript "UniFHy v0.1.1: A community modelling framework for the terrestrial water cycle in Python" presents a code framework for the coupling of distributed large scale models of the hydrological cycle – at least, this is how I understand the manuscript.

**AR:** No, this is a framework which allows decomposing land surface models developed for Earth System Models into only a few components. The proposed components correspond to ontologically well-defined entities. As they also correspond to disciplinary divisions, clear definitions of interfaces are essential so that microscale meteorologists understand the needs of hydrologists, or soil biochemists. This also explains why the subdivision of the land system needs to be as parsimonious as possible.

**R3C2:** The scope and the topic of the manuscript is not well defined and the basic motivation to create the code is not well explained. The introduction mentions a couple of existing modular modelling approaches but they do not really categorize the existing approaches and, most important, the introduction misses a section explaining what problems are not solved with the status quo and which benefits are created by the presented software.

**AR:** The paper in its introduction explains the limits of "existing modular modelling" in Earth system dynamics and in Earth system modelling communities. We argue that an intermediate approach is needed. UniFHy demonstrates that such a coarser subdivision of the land system, suitable for Earth system models, is achievable.

**R3C3:** From the whole manuscript I understand that UniFHy is a coupling interface definition, like OpenMI, OASIS-MCT / Open-PALM etc. that does not include solvers and / or model equations like the cited modular Hydrology tool kits (FUSE, SuperFLEX, CMF, SUMMA and RAVEN) or LandLab. However, the interface is extremely limited in scope and it is not well

explained, why such an interface is needed. My own experience is that any Python compatible API to model internals is sufficient for effective code coupling.

**AR:** It is exactly the ambition of the proposed interfaces to be limited in scope as they need to be implementable in OpenMI and OASIS. The solvers are internal to the three components proposed. It has to be recognised that a LSM is typically applied globally so that a large number of special cases need to be dealt with, thus general solvers are often unfit for this purpose.

**R3C4:** Discussions about model boundaries (eg. is root distribution subsurface or vegetation model component) is the most important part of model coupling from my experience. This discussion is missing in the manuscript and hindered by the interface. For my coupling studies such an interface would have been a hindrance and not an asset – even if Landsystem models are involved (CLM, LPJ-GUESS) (Bendix et al., 2021).

**AR:** Again this is internal to each of the three components proposed. The proposed interface can be implemented, and already partially exists in the LSMs the authors have contributed to (JULES, ORCHIDEE).

**R3C5:** The manuscript needs a complete rewrite to address the question, what unifhy really provides, what the status quo is missing. Large parts of the manuscript can be omitted (future development, case study in its current form), or need a complete rewrite focusing on the main questions (introduction, description of the framework, usage). I would therefore recommend to reject the manuscript in its current form.

**AR:** We believe that the clarifications brought to the introduction better places the objective of this paper within the state of the art. It should also better explain the innovation UniFHy provides in this effort to decompose LSMs into better defined components.

**R3C6:** The major problem is: the authors have not really decided which of two possible papers they want to write:
a) if the m/s is about the interface only, the need and the practical features of the interface and existing helper functions (eg. grid transformation, parallel memory allocation) should be presented. The status quo needs to be presented in a structured form
b) the presentation of the case study is to short and limited for a story of its own: The results are not discussed and the framework in its current form can be used for more variants then just the three presented.

The m/s is a mix of both papers, but none of the versions is presented in adequate form.

**AR:** This manuscript is presenting a new modelling framework. To this end, we believe that the reader needs to get a description of it, but also a demonstration that it actually can be used for what it is designed to do. This is precisely our intention with the case studies, and as mentioned by the reviewer, they are not meant to be a story of their own, this is why we are trying to keep it relatively short.

The framework is designed to allow the decomposition of the terrestrial water cycle into components such that (a) alternative science components can be compared, and that (b) varied spatial and temporal resolutions can be used for each component. To demonstrate this, the case studies need to show the framework outputs using (a) different modelling components, and (b) different resolutions (hence 3 configurations). In addition, we believe that it is important to demonstrate that it can work in more than one location (hence three study catchments). This requires a brief description of the components and configurations used.

But, as mentioned in the manuscript *"This brief analysis of the results is used to demonstrate the potential of the framework to elicit the most suitable combination of components to simulate the hydrological behaviour of a given region"* (see P17L303-P18L305 in the reviewed manuscript). Showcasing outputs (hydrographs and spatialised streamflow simulations) appears to us as the bare minimum in view to demonstrate that it functions, however, if the reviewer would like for the comparative analysis with the performance metrics to be removed from the analysis, we are happy to consider it.

**R3C7:** There are also code quality problems, like the use of underscore as a prefix in the public (user / contributor facing) interface.

**AR:** It is correct that the use of leading underscores is a convention in the Python community to denote "private" attributes of objects. As the reviewer points out, there are two levels of interaction with the framework: as a user (to run simulations) and as a contributor (to develop modelling components). On the one hand, users are not made aware nor expected to interact with "private" attributes, as per the convention. On the other hand, contributors are expected to assign values to "private" attributes to declare their modelling component's inputs/parameters/constants/outputs of their own classes. They inherit from framework component classes, and as such, it is not going against the convention to assign values to their own component class' "private" attributes. These attributes are made "private" following Python's convention so that users do not interact with them, which would most certainly result

in breaking the modelling component altogether. If the reviewer has other code quality problems to raise, we would be willing to consider them.

**Specific issues**

*1 Introduction*

**R3C8:** L40: CMF is not pure bucket style, but allows a wide range between physical and conceptual models – more physics than RAVEN (eg. Richards equation).

**AR:** Thank you for the correction. Given that the distinction in this paragraph was intended more about the granularity of the framework's building blocks, we agree that CMF belongs more with SUMMA and Raven as it is based on a finite volume method where process equations can indeed be replaced, as they form the model building blocks. The paragraph was rearranged accordingly (see P2L47-P3L52 in the revised manuscript).

**R3C9:** L43: "However, refactoring existing land surface models using these frameworks is not trivial" – citation needed. Counter examples for CMF coupling with CLM and LPJ-GUESS (Bendix et al., 2021). Using the proposed interface would be harder.

**AR:** This statement is made based on our experience, and experience collected from others (e.g. refactoring of CLM in SUMMA), so no particular reference can be included here. However, the sentence was rephrased to highlight that it is based on the authors' experience (see P3L52-43 in the revised manuscript).

**R3C10:** L44: OpenMI is something completely different from the modular hydrology toolboxes, it is more like the following paragraph

**AR:** As highlighted in the abstract, the framework is intended as a cross-community initiative, so we intentionally structured the existing literature per community of origin. Nevertheless, we do agree with the reviewer that OpenMI is closer to Earth System modelling frameworks in its philosophy, yet it was developed by hydro(geo)logists.

**R3C11:** L54: What is wrong with the status quo? Where are the gaps? Why don't you use an existing coupler or none at all?

**AR:** Current LSMs lack modularity and the entry-level to contribute to them is rather high and model-specific, which impedes comparisons of alternative processes from different LSMs. Frameworks used by hydrologists and Earth system dynamics are not fit for purpose because LSMs are components of ESMs and inherit the technical and computational requirements of ESMs which are not met by these frameworks. Existing modelling frameworks used for assembling ESMs feature such requirements, but they lack some specific needs of LSMs such as supermeshing or lateral fluxes crucial for hydrological processes. A new paragraph has been added to answer those questions (see P3L72-79 in the revised manuscript).

**R3C12:** L63: What is the "integrated coupling philosophy"? Which of the mentioned modular modelling approaches follow that also? What are the specific features of UnifHy?

**AR:** This term is explained earlier in the paragraph listing the existing efforts in the Earth system modelling communities (see P2L50-51 *"integrated coupling frameworks such as ESMF (Collins et al., 2005) or CPL7 (Craig et al., 2012), where existing modelling components require code refactoring to comply with a set of organising and interfacing requirements"* in the reviewed manuscript). ESMF and CPL7 are examples following this approach. UniFHy is an application of this existing approach to a new domain (i.e. the land system) with new interfacing requirements tailored to hydrology in the land system.

**2 Description of the framework**

**R3C13:** L85ff: Citation needed - why do you think a one size fits all solution ("skilfully yet pragmatically") exists? The "intentionally limited" "degrees of freedom offered" makes the proposed interface in my eyes anything else than "unified", but rather specialized for the purpose of this team and the models used in this study. What about energy or solute fluxes?

**AR:** The interfaces presented in the manuscript are the fruit of a large consultation with the land surface and hydrological modelling communities during workshops and follow-up discussions that will be summarised and made available separately (manuscript in preparation). As such, these interfaces are indeed specialised in subdividing the land system into finer components, based on the scope of such models and the processes typically taken into account in existing land surface models (e.g. ORCHIDEE, JULES, CLM). The models used in the case studies (i.e. Artemis, RFM, SMART) were subdivided as a subsequent step. The energy cycle is implicitly considered with the water cycle, even though thermodynamic fluxes are currently missing from the interface. The nutrient cycle is on the roadmap and is currently being defined separately.

**R3C14:** L98ff: Again, the term "integrated coupling approach" remains unexplained. The rationale for this structure is not explained

**AR:** See the author's reply to **R3C11**.

**R3C15:** L136: "These contributions can be implemented purely in Python, but can also rely on Fortran, C, or C++ programs called by interface middleware." From my experience, creating any interface is far from trivial. Using any interface for coupling is simple.

**AR:** We agree that interfacing compiled libraries with the Python framework is not trivial, but this does not make our statement any less valid. Indeed, the unit tests implemented in UniFHy do feature Fortran and C++ components wrapped in Python classes, which demonstrates that it is possible, and provides examples to follow in turn (i.e. using `numpy.f2py` and `cython`). The reader is made aware of this potential source for inspiration (see P15L256-257 in the revised manuscript).

**R3C16:** L154: "Contributors need not handle basic functionalities such as memory allocation nor input/output operations, as these are handled by the framework" Is this the main service of the provided software? How does the framework handle this? Why are the functions for memory allocation and input/output operations not explained? What are the limitations of the framework as is?

**AR:** As a general rule, given that the objective is to limit the burden of these tasks for the component contributors so that they can focus on the science aspects of the component, we are unsure of the relevance of documenting these aspects explicitly. That being said, regarding input/output operations, the contributor (and the manuscript reader) is made aware that the user is to provide NetCDF files whose data and metadata is following the CF conventions. Regarding memory operations, there is currently nothing special or innovative about our implementation worth mentioning, beyond the obvious care taken in avoiding unnecessary copies, memory leaks, and limiting the amount of data loaded in memory at once (through time slices). However, when the framework becomes ready to be used on distributed memory architectures (i.e. when domain parallelisation is implemented and compatible with the supermeshing approach), we do agree with the reviewer that the approach will be worth detailing.

*3 Usage of the framework*

**R3C17:** The whole section does not really follow a user story and is hard to follow. The code examples and descriptions seem to follow an example setup that is not yet described. The code examples lag explanations and comments.

**AR:** This section is structured as a step-by-step workflow that a user needs to follow, and as such we do believe that it follows a user story. Nonetheless, we have added a figure to provide an overview of the workflow to follow to set up and use the framework. We have also added paragraphs at the beginning of the section to better explain the content of the section (see P8L187-P9L194 in the revised manuscript), and more references between the text and the code examples. Together, we hope that these changes will contribute to make it easier to follow this section. As to the lack of code comments and explanations, please refer to the captions of the code snippets that we have expanded where appropriate.

**4 Contribution to the framework**

**R3C18:** This section would benefit of less code and more story.

**AR:** Providing less code appears undesirable as this would render the component invalid (i.e. it would not be a valid implementation accepted by the framework). So, we were only able to add a few extra elements to complete the story in this section.

**5 Case studies using the framework**

**R3C19:** The results are not discussed and the setup (and its rationale) is not well explained. Instead of focusing on the "unified" interface the m/s could present the results from this multi-model exercise as a scientific paper and mention, that more is possible using the interface.

**AR:** As already mentioned in the author's reply to **R3C6**, we believe that this section is required in this manuscript, and that is not meant to be a story of its own, but rather a demonstration of what is claimed previously. However, we agree with the reviewer that the rationale for the setup was lacking, so we have remedied this (see P15L280-284 in the revised manuscript). The results are already discussed in the reviewed manuscript (P16L293-P17L302), through some examples of comparisons of the performances of alternative configurations rather than a complete analysis of all configurations in all study catchments, which would be beyond the

scope of this manuscript, and which would go against the request from the reviewer not to try to produce a two-headed paper (as expressed in **R3C6**).

**6 Future developments**

**R3C20:** This section is far too long for things not yet existing. A short outlook is appropriate but two manuscript pages concerning not yet existing features is clearly too much – the future is always the most difficult time to predict, as we modellers know.

**AR:** As recommended by the reviewer, this section was removed, and the short outlook suggested is appended at the end of the Conclusions section (see P21L354-P22L366).

**References**

*Bendix, J., Aguire, N., Beck, E., Bräuning, A., Brandl, R., Breuer, L., Böhning-Gaese, K., de Paula, M. D., Hickler, T., Homeier, J., Inclan, D., Leuschner, C., Neuschulz, E. L., Schleuning, M., Suarez, J. P., Trachte, K., Wilcke, W., Windhorst, D., and Farwig, N.: A research framework for projecting ecosystem change in highly diverse tropical mountain ecosystems, Oecologia, 195, 589–600, https://doi.org/10.1007/s00442-021-04852-8, 2021.*

---

## Author Response (AR3)

Dear Dr. Dan Lu (Topical Editor), Dear Referee,

We would like to thank you for accepting our manuscript for publication in GMD. We are really grateful to the Editor for their time and for their work handling the various revisions of the manuscript.

The final manuscript version includes the following changes:
- as requested by the latest reviewer, we have removed the missing reference in line 149 that referred to the section on future work that no longer exists, and we replaced it with one example (it reads as "e.g. unstructured grids");
- we have adjusted the current affiliation for the first author;
- We have amended the acknowledgement section to thank the Editor and Reviewers.

On behalf of all authors,

Best wishes,
Thibault Hallouin